# *Leptospira* seroprevalence and associated risk factors among slaughterhouse workers in Western Bahr El Ghazal State, South Sudan

**David Onafruo**[1,2]*, **Anou Dreyfus**[3], **Joseph Erume**[4], **Clovice Kankya**[1], **Ambrose Jubara**[2], **Ikwap Kokas**[5], **Terence Odoch**[1], **Musso Munyeme**[6], **Lordrick Alinaitwe**[7], **Estella Kitale**[1,2], **Peter Marin**[1,8], **Esther Sabbath**[1,2], **Jörn Klein**[9,10]

1 Department of Biosecurity, Ecosystem and Veterinary Public Health, Makerere University, Kampala, Uganda, 2 Department of Clinical Studies, University of Bahr El Ghazal (UBG), Wau, South Sudan, 3 Section of Epidemiology, Vetsuisse Faculty, University of Zurich, Zürich, Switzerland, 4 Department of Biotechnical and Diagnostic Sciences, Makerere University, Kampala, Uganda, 5 Department of Biomolecular Resources and Biolab Sciences, Makerere University, Kampala, Uganda, 6 Department of Disease Control, University of Zambia, Lusaka, Zambia, 7 Central Diagnostic Laboratory, Makerere University (MU), Kampala, Uganda, 8 Department of Public Health, University of Bahr El Ghazal (UBG), Wau, South Sudan, 9 Department of Nursing and Health Sciences, Faculty of Health and Social Sciences, University of South-Eastern (USN) Porsgrunn, Norway, 10 Department of Microsystems, Faculty of Technology, Natural Sciences and Maritime Sciences, University of South-Eastern (USN), Porsgrunn, Norway

* donafruo@gmail.com

## Abstract

### Background

Leptospirosis is a neglected re-emerging and occupational zoonotic disease worldwide. In Africa, contact with livestock is postulated as a potential source of environmental contamination and a source of human *Leptospira* exposure, though pathways remain unknown. Recently, we confirmed *Leptospira* exposure and shedding among slaughtered cattle in Western Bahr El Ghazal. In the current study, we sought to determine corresponding occupational leptospiral seropositivity, associated risk factors and the prevalence of febrile illness among slaughterhouse workers.

### Methods

Between 27th February and 30th March 2023, we collected blood and interviewed 250 consenting slaughterhouse workers of the same facilities from which the cattle samples were collected. The workers were screened for leptospiral antibodies using the Microscopic agglutination test (MAT), based on a panel of 12 including those previously reported in livestock in South Sudan and the East African Region.

### Results

Of the 250 participants, 16 were seropositive 16/250, (6.4%, 95% CI = 3.2–10.2). Two seropositive individuals 0.8% (2/250) had MAT titers ≥ 800, indicative of probable recent leptospiral infection. Moreover, 42.4% (106/250) of the respondents reported experiencing fever in the past one month and 36.0% (90/250) sought medical attention. Among those seeking

**Data Availability Statement:** All data are available in the supplementary information

**Funding:** The study was funded by the Norwegian Agency for Development Cooperation (NORAD) through the NORHED II Program (Grant number 68802; https://en.uit.no/project/onehealth-cidimoh), received by CK, and implemented through the project Climate Change and Infectious Diseases Management - A One Health Approach (CIDIMOH). The funders had no role in study design, data collection and analysis, decision to publish, or preparation of the manuscript. None of the coauthors receive salaries from the funder.

**Competing interests:** The authors have declared that no competing interests exist.

medical care for febrile illnesses, diagnostic tests revealed 9.2% (23/250) with malaria, 7.6% (19/250) with typhoid, 16.8% (42/250) with both malaria and typhoid coinfections, 1.6% (4/250) with brucellosis, and no cases of leptospirosis had been considered. Most seropositive individuals reacted to serovar *L. borgpetersenii* Tarassovi 2.4% (6/250) and *L. interrogans* sv Australis 2.4% (6/250). The factors associated with seropositivity included flaying, with persons who flay animals having 14.9 times, (95% CI, 2.5–88.9) greater odds of being seropositive than persons who do not flay animals (P = 0.003), people who wore an apron/overall were 10.6 times (95% CI, 1.6–67.6) more likely to be seropositive than people who did not wear an apron/overall (P = 0.012). An increase in the number of carcasses handled per day by one increases the odds of exposure by 2.7 times (95% CI, 1.6–4.5), (P = 0.001).

## Conclusion

Finding seropositive workers in cattle slaughter facilities in Western Bahr El Ghazal, South Sudan, and similar serogroups as previously found in the cattle slaughtered at these facilities implies activities like animal slaughter that bring humans into close contact with animals could be one pathway for human *Leptospira* exposure in South Sudan. This could also highlight leptospirosis as a potential public health threat to those in frequent contact with cattle, including farmers, those in animal transportation, and veterinarians. The role of leptospirosis in cases of undifferentiated fever in South Sudan may also be worth investigating, especially in cases where occupational exposure is suspected. Further research including animals, the general public, farmers, and hospitalized patients is proposed to fully understand the burden of human leptospirosis. Including the serovar and serogroup Tarassovi and Australis in future vaccine development and serodiagnostic panels for South Sudan is highly recommended.

## Author summary

Leptospirosis is a re-emerging zoonotic disease affecting people who work closely with animals. In Africa, livestock contact is thought to be a potential source of human *Leptospira* exposure. This study examined occupational exposure among slaughterhouse workers in Western Bahr El Ghazal, South Sudan. From February 27 to March 30, 2023, blood samples and interviews were collected from 250 slaughterhouse workers. The workers were tested for leptospiral antibodies using the Microscopic Agglutination Test (MAT).

Results showed that 6.4%, 16/250 of the workers were seropositive for *Leptospira*, with two individuals showing high antibody levels indicating recent infection. Additionally, 42.4% reported having had a fever in the past one month. Seropositive individuals mostly reacted to *L. borgpetersenii* Tarassovi and *L. interrogans* Australis. Factors linked to higher seropositivity included flaying animals, wearing protective clothing, and handling more carcasses per day.

The study concludes that contact with cattle in slaughter facilities is a significant pathway for human leptospiral exposure, underscoring leptospirosis as a public health risk for those frequently in contact with cattle, including farmers and veterinarians. This highlights the need for further investigation into leptospirosis in cases of undifferentiated fever for workers presenting with fever in South Sudan. Subsequent research should involve

hospitalized patients to gain a complete understanding of the disease's burden on humans, together with identifying local reservoir hosts. We recommend the inclusion of serogroups Tarassovi and Australis in future vaccine development and serodiagnostic panels for South Sudan.

## Introduction

Leptospirosis represents a frequently overlooked, re-emerging, and occupationally relevant zoonotic disease that has a significant public health impact worldwide, with little data available from Africa [1–3]. This impact is particularly critical within economically vulnerable populations of developing countries like South Sudan. Slaughterhouse workers are among the high-risk groups [4,5]. The global incidence of leptospirosis is estimated at 1.03 million cases and 60,000 deaths annually, with case fatality ranging from < 5% to 70% [3,6]. The highest average annual incidence of leptospirosis is in Africa, estimated at 95.5 per 100,000 people [7].

The causative agent of leptospirosis is pathogenic Gram-negative bacteria of the genus *Leptospira*, comprising 69 species and over 300 serovariants/serovars [8]. Unlike in animals where certain *Leptospira* serovars are reportedly adapted, humans are susceptible to infection by a diverse range of pathogenic *Leptospira* species [9,10]. *Leptospira* transmission to humans usually occurs through direct exposure to urine, body fluids, or aborted tissues of infected animals or from indirect contact with water or soil that has been contaminated with the urine of infected animals [11–14].

In Africa, contact with livestock is postulated as a potential source of environmental contamination and a source of human *Leptospira* exposure. However, few data are available to allow comparison of *Leptospira* infection in linked human and animal populations. Only one study on human leptospirosis can be found in South Sudan, reporting 0.95% (6/632) cases among Mongolian peacekeepers deployed to South Sudan for 9 months [15]. Recently we confirmed leptospiral seropositivity and shedding among slaughtered cattle in Western Bahr El Ghazal. [16]. In the current study, we sought to determine corresponding occupational leptospiral seropositivity, associated risk factors and the prevalence of febrile illness among slaughterhouse workers in these slaughter facilities from which we sampled the cattle.

## Materials and methods

### Ethics statement

The study procedures involving sample collection from human subjects were approved by the Ministry of Health, Research Ethics Review Board (MOH-RERB), Juba, South Sudan, (MOH/RERB 02/7/2022—MOH/RERB/A/02/6/02/2023), the recruited participants were informed of the study objectives before and during sample collection, each participant provided written informed consent to participate in the study. For participants below 18 years, verbal assent from the participant and formal written consent was obtained from the parent/guardian for child participants as the legal age for consent to medical procedures in South Sudan is 18 years.

### Study site, study design, and study population

This cross-sectional study was conducted in the Wau Municipal Council, Western Bahr El Ghazal State, South Sudan, from February 27th to March 30th, 2023. Wau Municipal Council was selected for the study because it is the largest town in the greater Bahr El Ghazal region,

serving as a central hub for livestock marketing. Additionally, the region's largest slaughterhouse has an average of 50 cattle slaughtered daily and is staffed with experienced veterinarians.

Participants for the study were recruited from three major slaughter facilities, namely Lokoloko, Zagalona, and Eastern Bank. The facilities slaughter cattle, sheep, and goats, and we had previously confirmed *Leptospira* shedding among the cattle slaughtered in these facilities around the same time we sampled the workers [16].

## Sample Size estimation, sampling strategy

The sample size of 193 workers was estimated in epitools epidemiological calculators [17], and based on a leptospiral seroprevalence of 0.95% (6/632) previously reported among peacekeepers deployed in South Sudan in 2012 [15] and considering an imperfect MAT test with sensitivity and specificity of 55% and 97%, respectively [18], a 95% confidence level and precision of 0.05. The estimated sample was oversampled to 250 to cover for possible problems with non-response or missing values [19].

## Sample and data collection

Before sample collection, meetings were held with slaughterhouse workers to explain study objectives and procedures. Workers who consented to participate had a study nurse take about 5mL of venous blood sample collected into a gel-activated serum tube.

Additionally, the participants were interviewed using a pre-tested digital questionnaire designed in open data kit (ODK version 2021.2.3). The first author administered the interviews in English and Arabic with the help of pre-trained translators for participants who did not speak English or Arabic. The interview captured information on the workers'demography, risk factors for exposure at work [number of months working in the slaughterhouse, working days per month, working hours per day, incidents such as animal blood and/ urine splashed in eyes or mouth or cuts on hands or legs in the last three months, personal protective and involvement in other equally risky activities such as hunting, farming, fishing, handling aborted materials, and their recent medical history. Work positions were not considered since these were not specialized (Each worker participated in all steps in the slaughter process).

## Sample processing and storage

Samples were kept on ice in a cooler box during collection, and serum was prepared on arrival at the Wau Teaching Hospital's Expanded Program on Immunization (EPI) following centrifugation at 5000rpm (Harvard, Eltek Labspin TC 450 C) at room temperature for 5 minutes. The resulting sera were aliquoted into 2 mL cryogenic vials and stored at -20˚C. Subsequently, the sera were transported by air from South Sudan to Uganda, Central Diagnosis Laboratory (CDL) at Makerere University, College of Veterinary Medicine, Animal Resources, and Biosecurity (COVAB), where they were stored at -20˚C until laboratory testing.

**Serological testing.** The microscopic Agglutination Test (MAT), recognized as the gold standard, was utilized to distinguish *Leptospira* spp. serogroups antibodies in the serum samples of humans as described by OIE standards [20]. A panel of 12 serovars (sv) belonging to 12 serogroups (sg) and four *Leptospira* spp. (**S1 Table**) were sourced from the Academic Medical Center (Leptospirosis Reference Center), the Netherlands and used as live test antigens. The panel was selected to include serovars reported as predominant in Sudan [21], and East Africa [22–25].

The sera samples were screened at an initial dilution of 1:50. The positive reacting samples were then further titrated in 2-fold dilution to determine the endpoint titer, defined as the

highest serum dilution capable of agglutinating $\geq$ 50% of leptospires [26]. Sera with a titer $\geq$ 100 against any *Leptospira* spp. serovar were considered seropositive [27]. Commercial rabbit antisera against each serovar on the panel was included as positive controls and normal saline as the negative control.

## Data analysis

Questionnaire information was retrieved from ODK and subsequently transferred to Microsoft Excel 2016 (Microsoft Corp, Redmond WA, USA) for data cleaning and coding processes. Similarly, serological test results were entered into Microsoft Excel, and all the data underwent analysis using SPSS software (IBM SPSS statistics version 26). Exploratory data analysis was conducted to validate the data and evaluate crude associations by using Chi-Square Tests, 2 × 2 tables, and summary measures.

The outcome variables included the overall seroprevalence of respondents testing positive for any serovar, specific serovars, probable recent leptospirosis, and an examination of cross-reactivity among different serovars (sv) and serogroups (sg). To assess past medical history, we evaluated instances of seeking medical care, disease diagnostic results, and symptoms based on the last one-month before the blood sample was taken. Univariable logistic regression first explored the associations between overall seroprevalence and various exposure variables. Additionally, multivariable logistic regression analysis, employing the manual backward Wald method, was utilized to ensure model diagnostics/assumptions. This method aids in examining associations between the remaining exposure variables and *Leptospira* spp. seropositivity, thereby controlling for other variable effects. Exposure variables were entered into the model if the univariable p-value was $\leq$ 0.2 and retained if the likelihood ratio test was statistically significant (p $\leq$ 0.05). Variables were excluded if the univariable p-value was > 0.2. We also checked for multicollinearity using the variance inflation factor (VIF), and variables with VIF > 5 were considered and excluded from the model.

**Case definitions.** A seropositive case was a participant with a MAT titer of $\geq$ 100 against any *Leptospira* spp. serovar.

A probable leptospirosis case was a participant with a MAT titer of $\geq$ 800 against any *Leptospira* spp. serovar [6,25,28].

## Results

### Worker demography and Leptospira seroprevalence

A total of 263 slaughterhouse workers participated in the initial recruitment meetings, 250 of whom consented to participate in the study. Up to 71.2% (178/250) of the workers sampled were from Lokoloko slaughterhouse, 10.8% (27/250) from Zagalona slaughterhouse and 18% (45/250) were from Eastern Bank. Eighty percent of the participants (200/250) were male. Moreover, 42.4% (106/250) of the workers reported having experienced fever in the month before sampling, and five of them (4.7%, 5/106, 95% CI, 0.9–9.6) were seropositive **Table 1**.

Leptospiral antibodies MAT titer of $\geq$ 100 were detected in 16 of the 250 workers (6.4%, 95% CI: 3.2–10.2). Only five serovars reacted positively with the serum samples: *L. borgpetersenii* sv. Tarassovi, *L. interrogans* sv Australis, *L. kirschneri* sv Grippotyphosa, *L. borgpetersenii* sv Keny and *L. interrogans* sv Hebdomadis **Tabel 2**. None of the samples reacted to the other serovars in the panel (S2 Table). Among the seropositive workers, the majority showed reactivity to *L. borgpetersenii* sv. Tarassovi (2.4%, 6/250), and *L. interrogans* sv Australis (2.4%, 6/250). Two workers (0.8%, 2/250) had high leptospiral antibody titres $\geq$ 800, suggesting probable acute *Leptospira* infection **Table 2**.

**Table 1.** Demographic Characteristics of the Study Population (*N* = 250), and association with leptospiral seropositivity.

| Exposure Variables | Categories | n (%) | Seropositivity (%) | X² P-Value | 95% C.I. | |
|---|---|---|---|---|---|---|
| | | | | | Lower | Upper |
| Slaughter facility | Lokoloko | 178(71.2) | 13(7.3) | 0.115 | 3.3 | 12.6 |
| | Zagalona | 27(10.8) | 3(11.1) | | 0.0 | 24.0 |
| | Eastern Bank | 45(18) | 0(0.0) | | - | - |
| Sex | Men | 200(80) | 16(8.0) | 0.039 | 4.5 | 11.9 |
| | Women | 50(20) | 0(0.0) | | - | - |
| Age | 14–19 | 19(7.6) | 2(10.5) | 0.621 | 0.0 | 28.6 |
| | 20–29 | 78(31.2) | 6(7.7) | | 2.4 | 13.6 |
| | 30–39 | 77(30.8) | 4(5.2) | | 1.2 | 10.3 |
| | 40–49 | 53(21.2) | 4(7.5) | | 0.0 | 16.2 |
| | >50 | 23(9.2) | 0(0.0) | | - | - |
| Religion | Christian | 159(63.6) | 9(5.7) | 0.528 | 2.5 | 9.5 |
| | Muslim | 91(36,4) | 7(7.7) | | 3.3 | 13.5 |
| Educational level | No formal level | 50(20) | 3(6.0) | 0.932 | 0.0 | 12.5 |
| | Primary level | 122(48.8) | 7(5.7) | | 1.8 | 10.9 |
| | Secondary level | 61(24.4) | 5(8.2) | | 1.8 | 15.4 |
| | Tertiary level | 17(6.8) | 1(5.9) | | 0.0 | 22.0 |
| Attended training | No | 196(78.4) | 11(5.6) | 0.332 | 2.1 | 9.1 |
| | Yes | 54(21.6) | 5(9.3) | | 1.9 | 17.0 |
| Wearing apron/overall | No | 122(48.8) | 3(2.5) | 0.013 | 0.0 | 5.3 |
| | Yes | 128(51.2) | 13(10.2) | | 5.3 | 15.7 |
| Use of abattoir attires | No | 147(58.8) | 13(8.8) | 0.059 | 4.6 | 14.1 |
| | Yes | 103(41.2) | 3(2.9) | | 0.0 | 6.7 |
| Flaying | No | 214(85.6) | 6(2.8) | 0.001 | 0.9 | 5.4 |
| | Yes | 36(14.4) | 10(27.8) | | 10.7 | 44.1 |
| Meat cutting | No | 189(75.6) | 10(5.3) | 0.207 | 2.1 | 8.9 |
| | Yes | 61(24.4) | 6(9.8) | | 2.9 | 18.6 |
| Number of carcasses handled per day | 1 | 70(28.0) | 1(1.4) | 0.002 | 0.0 | 4.5 |
| | 2 | 37(14.8) | 0(0.0) | | - | - |
| | 3 | 21(8.4) | 2(9.5) | | 0.0 | 23.7 |
| | 4 | 16(6.4) | 4(25.0) | | 6.0 | 50.0 |
| | 6 | 84(33.6) | 9(10.7) | | 4.8 | 17.6 |
| Working hours per day | 1 | 1(0.4) | 0(0.0) | 0.573 | - | - |
| | 2 | 78(31.2) | 3(18.8) | | 0.0 | 36.5 |
| | 3 | 96(38.4) | 6(37.5) | | 17.8 | 66.0 |
| | 4 | 75(30.0) | 7(43.8) | | 19.3 | 69.1 |
| Past Medical History ≤ a month | Felt sick | 192(76.8) | 10(5.2) | 0.161 | 2.1 | 8.6 |
| | Went to hospital | 90(36.0) | 3(3.3) | | 0.0 | 7.1 |
| | Fever | 106(42.4) | 5(4.7) | | 0.9 | 9.6 |
| | Malaria | 23(9.2) | 1(4.3) | | 0.0 | 14.7 |
| | Typhoid | 19(7.6) | 1(5.3) | | 0.0 | 19.6 |
| | Malaria and Typhoid | 42(16.8) | 2(4.8) | | 0.0 | 13.3 |
| | Brucellosis | 4(1.6) | 0(0.0) | | - | - |
| Source of drinking water | | | | | | |
| Tap/Pipped | No | 210(84.0) | 13(6.2) | 0.756 | 2.8 | 10.0 |
| | Yes | 40(16.0) | 3(7.5) | | 0.0 | 17.2 |

(*Continued*)

**Table 1.** (Continued)

| Exposure Variables | Categories | n (%) | Seropositivity (%) | X² P-Value | 95% C.I. | |
|---|---|---|---|---|---|---|
| | | | | | Lower | Upper |
| Borehole | No | 193(77.2) | 14(7.3) | 0.310 | 4.0 | 10.9 |
| | Yes | 57(22.8) | 2(3.5) | | 0.0 | 9.4 |
| Open wells | No | 31(12.4) | 0(0.0) | 0.120 | - | - |
| | Yes | 219(87.6) | 16(7.3) | | 4.1 | 10.4 |
| Seeing rats | Daily | 195(78.0) | 12(6.2) | 0.910 | 2.6 | 10.2 |
| | At least once a week | 30(12.0) | 3(10.0) | | 0.0 | 20.9 |
| | At least once in a month | 2(0.8) | 0(0.0) | | - | - |
| | Not seen in a year | 1(0.4) | 0(0.0) | | - | - |
| | Never seen | 22(8.8) | 1(4.5) | | 0.0 | 19.0 |
| Overall *Leptospira* spp. seroprevalence | Any serovar | 250(100) | 16(6.4) | | 3.2 | 10.2 |

X² = Chi-Square Tests

## Risk factors for *Leptospira* seroprevalence

In the univariable analysis, *Leptospira* seropositivity was associated with flaying, the proxy of wearing aprons/overalls, meat cutting, handling cattle, absence of abattoir attire use, the number of carcasses handled by an individual daily, and working hours at the slaughterhouse were selected for univariable logistic regression analysis. The final multivariable regression model retained the variables flaying, wearing aprons/overalls, and the number of carcasses handled daily (P ≤ 0.05). Flayers had 14.9 times (95% CI, 2.5–88.9, P = 0.003) greater odds of being seropositive compared to those in other worker positions. Those who wore aprons or overalls while working had 10.6 times higher odds (95% CI, 1.6–67.6) greater odds of being seropositive than persons who did not wear (P = 0.012). An increase in the number of carcasses handled per day by one increases the odds of exposure by 2.7 times (95% CI, 1.6–4.5), (P = 0.001) **Table 3.**

## Discussion

In this cross-sectional study based in slaughterhouses, the first of its kind in the region and South Sudan, we investigated the seroprevalence of leptospiral antibodies in slaughterhouse

**Table 2. Leptospiral serovar, leptospiral seropositivity, determined by the Microscopic Agglutination Test (titer ≥ 100), among slaughterhouse workers (N = 250) sampled in Western Bahr El Ghazal State, South Sudan.**

| *Leptospira* spp. Serovar | MAT titer | | | | | | | | n Positive | Prevalence*, 95% CI |
|---|---|---|---|---|---|---|---|---|---|---|
| | 100 | 200 | 400 | 800 | 1600 | 3200 | 6400 | 12800 | | |
| *L. borgpetersenii* sv Tarassovi | 0 | 2 | 4 | 0 | 0 | 0 | 0 | 0 | 6 | 2.4% (0.4–4.4) |
| *L. interrogans* sv Australis*** | 2 | 2 | 1 | 1 | 0 | 0 | 0 | 0 | 6 | 2.4% (0.8–4.4) |
| *L. kirschneri* sv Grippotyphosa*** | 3 | 0 | 0 | 0 | 0 | 0 | 0 | 0 | 3 | 1.2% (0.0–2.8) |
| *L. borgpetersenii* sv Kenya | 0 | 1 | 0 | 1 | 0 | 0 | 0 | 0 | 2 | 0.8% (0.0–2.0) |
| *L. interrogans* sv Hebdomadis*** | 1 | 0 | 0 | 0 | 0 | 0 | 0 | 0 | 1 | 0.4% (0.0–1.2) |
| | 6 | 5 | 5 | 2** | 0 | 0 | 0 | 0 | 18*** | |
| Any *Leptospira* spp., serovar | Any positive excluding cross-reaction | | | | | | | | 16 | 6.4* % (3.2–10.2) |

* This is an apparent prevalence since the Microscopic Agglutination Test (MAT) is not 100% sensitive and specific.

**Probable recent leptospirosis MAT titer = 800

*** Out of the total of 16 positive sera, one cross–reacted simultaneously with the three serovars of three different serogroups.

**Table 3. Population characteristics, seroprevalence (%) of *Leptospira* spp. and associated risk factors analysed using univariable and multivariable logistic regression models of sampled slaughterhouse workers (*N* = 250) in the Western Bahr El Ghazal Region, South Sudan, during February and March 2023.**

| Variables | Categories | n (%) | Univariable | | | | Multivariable | | | |
|---|---|---|---|---|---|---|---|---|---|---|
| | | | OR | P value | 95% C.I. | | OR | P value | 95% C.I. | |
| | | | | | Lower | Upper | | | Lower | Upper |
| Slaughter facility | Lokoloko | 178(71.2) | Ref. | | | | | | | |
| | Zagalona | 27(10.8) | 1.5 | 0.792 | 0.4 | 5.9 | | | | |
| | Eastern Bank | 45(18) | 0.0 | | - | - | | | | |
| Sex | Men | 200(80) | Ref. | | | | | | | |
| | Women | 50(20) | 0.0 | 0.997 | - | - | | | | |
| Age | 14–19 | 19(7.6) | Ref. | | | | | | | |
| | 20–29 | 78(31.2) | 0.7 | 0.938 | 0.1 | 3.8 | | | | |
| | 30–39 | 77(30.8) | 0.4 | | 0.0 | 2.7 | | | | |
| | 40–49 | 53(21.2) | 0.6 | | 0.1 | 4.1 | | | | |
| | >50 | 23(9.2) | 0.0 | | - | - | | | | |
| Religion | Christian | 159(63.6) | Ref. | | | | | | | |
| | Muslim | 91(36,4) | 1.3 | 0.529 | 0.4 | 3.8 | | | | |
| Educational level | No formal level | 50(20) | Ref. | | | | | | | |
| | Primary level | 122(48.8) | 0.9 | 0.933 | 0.2 | 3.846 | | | | |
| | Secondary level | 61(24.4) | 1.3 | | 0.3 | 6.163 | | | | |
| | Tertiary level | 17(6.8) | 0.9 | | 0.0 | 10.0 | | | | |
| Attended training | No | 196(78.4) | Ref. | | | | | | | |
| | Yes | 54(21.6) | 1.7 | 0.337 | 0.5 | 5.1 | | | | |
| Wearing Aprons/overall | No | 122(48.8) | Ref. | | | | | | | |
| | Yes | 128(51.2) | 4.4 | 0.022* | 1.2 | 16.1 | 10.6 | 0.012** | 1.6 | 67.6 |
| Wearing gumboot | No | 180(72) | Ref. | | | | | | | |
| | Yes | 70(28) | 1.5 | 0.385 | 0.5 | 4.5 | | | | |
| Use of abattoir attires | No | 147(58.8) | Ref. | | | | | | | |
| | Yes | 103(41.2) | 0.3 | 0.073* | 0.0 | 1.1 | | | | |
| Flaying | No | 214(85.6) | Ref. | | | | Ref. | | | |
| | Yes | 36(14.4) | 13.3 | 0.001* | 4.4 | 39.7 | 14.9 | 0.003** | 2.5 | 88.9 |
| Meat cutting | No | 189(75.6) | Ref. | | | | | | | |
| | Yes | 61(24.4) | 1.9 | 0.214* | 0.6 | 5.6 | | | | |
| Number of carcasses handled per day | | 250(100) | 1.4 | 0.004* | 1.1 | 1.9 | 2.7 | 0.001** | 1.6 | 4.5 |
| Working hours per day | | 250(100) | 1.6 | 0.165* | 0.8 | 3.1 | | | | |
| Tap/Pipped | No | 210(84.0) | Ref. | | | | | | | |
| | Yes | 40(16.0) | 1.2 | 0.757 | 0.3 | 4.5 | | | | |
| Borehole | No | 193(77.2) | Ref. | | | | | | | |
| | Yes | 57(22.8) | 0.4 | 0.321 | 0.1 | 2.1 | | | | |
| Open wells | Yes | 219(87.6) | Ref. | | | | | | | |
| | No | 31(12.4) | 0.0 | 0.998 | - | - | | | | |
| Seeing rats | Daily | 195(78.0) | Ref. | | | | | | | |
| | At least once a week | 30(12.0) | 1.2 | 0.950 | 0.2 | 6.2 | | | | |
| | At least once in a month | 2(0.8) | - | | - | - | | | | |
| | Not seen in a year | 1(0.4) | - | | - | - | | | | |
| | Never seen | 22(8.8) | 1.4 | | 0.1 | 18.5 | | | | |

* Variables with P- value ≤ 0.2 in the univariable logistic regression models taken to multivariable logistic regression model.

** variables with statistically significant P- value in the multivariable logistic regression model.

workers. The outcomes support the hypothesis of potential endemic leptospirosis as a zoonotic disease among humans in the area and the country. Compared to a previous serosurvey of South Sudan UN Mongolian Peacekeepers, reported a 0.95% (6/632) seroconversion using a commercial ELISA [15], our findings revealed a significantly higher leptospiral seroprevalence of 6.4% (16/250). While acknowledging that the MAT and ELISA methods are not 100% comparable, this substantial difference in seroprevalence suggests a higher risk among slaughterhouse workers compared to the observed seroconversion in UN peacekeeping forces. The shorter exposure duration [averaging 9 months] for peacekeepers in South Sudan might have contributed to the lower seroconversion in that group and less exposure to cattle, which are strong shedders of *Leptospira* spp., particularly the emerging serovar and serogroup Tarassovi reported in South Sudan from the same slaughter facilities where workers were sampled [16]. This finding aligns with the strong association between shedding and seropositivity in cattle for sv. Tarassovi, posing a public health threat to dairy farmers in New Zealand and Thailand [29–31].

Although the difference in *Leptospira* spp., seroprevalence among the various slaughter facilities was not statistically significant, it is worth noting that, workers from the Eastern Bank slaughter facility 18% (45/250) had 0.0% leptospiral seropositivity. This particular slaughterhouse is located outside of town. The workers who perform the riskiest activities, such as slaying, flaying, and evisceration, did not participate in the study because they typically rush to the market after completing their slaughter work. Instead, the participants from this slaughterhouse were mostly dealers of offal, offcuts, skin, and live animal traders.

Our estimated leptospiral seroprevalence varies compared to different results reported elsewhere within East African region. In Uganda, the seroprevalence among healthcare patients was recorded at (126/359, 35%,95% CI, 30.2–40.3%) [25], in Tanzania at (7/41, 17.1%, 95%CI, 7.1–32.1%) among abattoir workers [32], and in Kenya at (99/737, 13.4%, 95%CI, 11.1–16.1%) among slaughterhouse workers [33]. The observed differences could be due to obvious different prevalence in host species in these countries, different exposure to transmission sources, such as host species, contaminated water sources and streams. Further, due to variations in sample size, seasonality, serovars included in the testing panels, and study design methods used. The predominant serovar Tarassovi, and sv. Australis were in agreement with findings reported among cattle in the Upper Nile Province in Sudan [21]. A similar finding was reported among slaughtered cattle in the Bahr El Ghazal region of South Sudan, where 78.6% (316/402) of cattle tested seropositive against sg. Tarassovi [16].

The high prevalence of antibodies against the livestock-associated sv. Tarassovi detected in this study hence suggests a potential risk of *Leptospira* spp. transmission between cattle and humans in the Bahr El Ghazal region of South Sudan. Findings from studies in New Zealand and Thailand further support this risk [29–31]. Further, there is also a possibility of a transmission dynamic between swine, cattle and humans, since pigs and cattle have been identified as potential reservoirs for sv. Tarassovi [21,32,34].

One of the significant risk factors associated with leptospiral seropositivity in the multivariable logistic regression model was animal flaying (OR 12.6). Flaying as a risk factor concords with the results reported from New Zealand among slaughterhouse workers [35]. A probable explanation is that animals spend nights in a confined area at the slaughterhouse, potentially increasing the risk of skin contamination with their urine. This might put those directly involved in flaying at higher risk compared to individuals engaged in other activities like meat cutting.

In our study, wearing an apron could be a proxy for doing activities in the slaughterhouse, where exposure to urine is possible. Leptospires can be transmitted by cuts, intact skin immersed in contaminated water with leptospires for a long period, and through mucous

membranes contact, hence wearing protective clothing/aprons might have a less protective role in case of leptospirosis [33,36]. Further comparative observational studies need to be done on work practices in settings where overall/protective gears are used and comparing them to the settings where overall/protective gears are not used.

Another risk factor associated with *Leptospira* spp. seropositivity was the number of carcasses handled per day, as the number of carcasses increases from one, the higher chance of exposure to *Leptospira* spp. increases by (OR) of 2.3 times compared to persons who handle one or fewer animals per day. This can be explained by the frequency of exposure which will increase the likelihood of getting in contact with an infected animal. Similar findings were reported in the case of *Brucella* spp. from South Korea among cattle slaughterhouse [37].

Our study revealed a high prevalence of febrile illness in slaughterhouse workers. While Brucellosis and Typhoid are being tested for, leptospirosis is not. We highly recommend including this pathogen in the diagnostic test panel for febrile slaughterhouse workers along with treatment guidelines for medical staff. It is further recommended to increase the awareness of zoonotic diseases and transmission pathways in slaughterhouse workers and introduce personal protective equipment (goggles, gloves etc.) together with training on how these should be used.

The study's findings are not representative of the *Leptospira* spp. seroprevalence status among humans in the region and we did not determine the risk of exposure to different livestock species as the study was conducted in multispecies slaughterhouses. Hence, future research should include seroepidemiological surveys covering the general public and farmers, along with identifying reservoir hosts. To enhance our holistic understanding of the disease burden in humans, integrating molecular diagnostic tools like PCR for investigating active infections among hospitalized febrile patients would significantly contribute to the knowledge base of the leptospirosis burden in South Sudan. This research was carried out among apparently healthy workers, and given the voluntary nature of participation, it's important to interpret the findings within the setting of the same study population. To address the limitation of a cross-sectional study design, which prevents establishing a causal relationship with leptospirosis, we propose further longitudinal studies for a more robust understanding of these associations. Another limitation of this study is the busy schedule of some slaughterhouse workers and their unwillingness to participate in the study, this may have introduced a selection bias if the busy schedule was associated with activities leading to exposure to leptospires. Slaughter workers are a risk population for leptospirosis infection and findings cannot be extrapolated to the general population.

## Conclusion

Detecting seropositive workers in cattle slaughter facilities in Western Bahr El Ghazal, South Sudan, suggests that activities like animal slaughter could be a pathway for human *Leptospira* exposure. This highlights leptospirosis as a potential public health threat, especially for those frequently in contact with cattle. Investigating its role in undifferentiated fever cases with suspected occupational exposure is crucial. Future efforts should include using molecular diagnostic tools like PCR, identifying local animal reservoirs, and expanding serodiagnosis panels to cover more serovars, including Tarassovi and Australis, for improved disease management and vaccine development.

## Supporting information

**S1 Table. Panel of *Leptospira* spp. serovars used as live antigens in the Microscopic Agglutination Test.**
(DOCX)

**S2 Table. Seroprevalence of *Leptospira* spp. serovars, serogroups and strains by Microscopic Agglutination Test.**
(DOCX)

## Author Contributions

**Conceptualization:** David Onafruo, Anou Dreyfus, Joseph Erume, Clovice Kankya, Ambrose Jubara, Ikwap Kokas, Terence Odoch, Musso Munyeme, Lordrick Alinaitwe, Estella Kitale, Peter Marin, Esther Sabbath, Jörn Klein.

**Data curation:** David Onafruo, Ikwap Kokas, Lordrick Alinaitwe, Estella Kitale, Peter Marin, Esther Sabbath.

**Formal analysis:** David Onafruo, Anou Dreyfus, Lordrick Alinaitwe, Peter Marin.

**Funding acquisition:** Clovice Kankya, Ambrose Jubara.

**Investigation:** David Onafruo, Estella Kitale, Peter Marin, Esther Sabbath.

**Methodology:** David Onafruo, Anou Dreyfus, Joseph Erume, Terence Odoch, Musso Munyeme, Lordrick Alinaitwe, Estella Kitale, Jörn Klein.

**Project administration:** Clovice Kankya.

**Resources:** Clovice Kankya.

**Software:** David Onafruo.

**Supervision:** Anou Dreyfus, Joseph Erume, Clovice Kankya, Jörn Klein.

**Validation:** David Onafruo, Anou Dreyfus, Joseph Erume, Clovice Kankya, Ambrose Jubara, Ikwap Kokas, Lordrick Alinaitwe, Jörn Klein.

**Visualization:** David Onafruo, Jörn Klein.

**Writing – original draft:** David Onafruo, Anou Dreyfus, Terence Odoch, Musso Munyeme, Lordrick Alinaitwe, Estella Kitale, Peter Marin, Jörn Klein.

**Writing – review & editing:** David Onafruo, Anou Dreyfus, Joseph Erume, Clovice Kankya, Ambrose Jubara, Ikwap Kokas, Terence Odoch, Musso Munyeme, Lordrick Alinaitwe, Jörn Klein.

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
