## [Decision Letter · Decision Letter 0]

24 Jun 2024

Dear Dr. Kal,

Thank you very much for submitting your manuscript "Seroprevalence and associated risk factors of Leptospira spp. among slaughterhouse workers in Western Bahr El Ghazal State, South Sudan" for consideration at PLOS Neglected Tropical Diseases. As with all papers reviewed by the journal, your manuscript was reviewed by members of the editorial board and by several independent reviewers. In light of the reviews (below this email), we would like to invite the resubmission of a significantly-revised version that takes into account the reviewers' comments. 

Please address all the comments from the reviewers before submitting the revised version. While revising Please clearly discuss the limitations of the study.

We cannot make any decision about publication until we have seen the revised manuscript and your response to the reviewers' comments. Your revised manuscript is also likely to be sent to reviewers for further evaluation.

Sincerely,

Sreekumari Rajeev, BVSc, PhD

Guest Editor

Ana LTO Nascimento

Section Editor

Please address all the comments from the reviewers before submitting the revised version. While revising Please clearly discuss the limitations of the study.

Reviewer's Responses to Questions

**Key Review Criteria Required for Acceptance?**

**Methods**

-Are the objectives of the study clearly articulated with a clear testable hypothesis stated?

-Is the study design appropriate to address the stated objectives?

-Is the population clearly described and appropriate for the hypothesis being tested?

-Is the sample size sufficient to ensure adequate power to address the hypothesis being tested?

-Were correct statistical analysis used to support conclusions?

-Are there concerns about ethical or regulatory requirements being met?

Reviewer #1: (No Response)

Reviewer #2: Yes, clear and straightforward design; appropriate sample size and tested population, standard lab and statistical procedures were used in the study.

**Results**

-Does the analysis presented match the analysis plan?

-Are the results clearly and completely presented?

-Are the figures (Tables, Images) of sufficient quality for clarity?

Reviewer #1: Yes

Yes

Yes

Reviewer #2: Yes, results are clearly presented.

**Conclusions**

-Are the conclusions supported by the data presented?

-Are the limitations of analysis clearly described?

-Do the authors discuss how these data can be helpful to advance our understanding of the topic under study?

-Is public health relevance addressed?

Reviewer #1: Yes

Yes-although I have recommended additional limitations be presented as well

Yes

Yes

Reviewer #2: Yes. Conclusions drawn are based on the results.

**Editorial and Data Presentation Modifications?**

Reviewer #1: Comments

Abstract > results

Is there something missing here? ‘no cases of leptospirosis were considered’ should it read ‘ no cases of leptospirosis had been considered’

Author summary

Consider rephrasing to say ‘presenting with fever’ instead of ‘presenting fever’

Introduction

Check reference formatting to make it uniform to the rest of the manuscript ‘leptospirosis is in Africa estimated at 95.5 per 100,000 people (Mgode et al., 2015).’

Discussion

Check this sentence for grammatical correctness. ‘handle at less’ sounds off. ‘Leptospira spp. increases by (OR) of 2.3 times compared to persons who handle at less one animal’

Reviewer #2: There is some redundancy in a few sentences, e.g., presence of leptospira antibody seropositivity (should be changed to leptospiral antibodies or leptospiral seropositivity)

**Summary and General Comments**

Reviewer #1: This well written manuscript outlines the prevalence of Leptospirosis among slaughter house workers in South Sudan. The manuscript conveys impactful information about this zoonotic disease in an area where little has been published about it and using non-random probability sampling techniques the authors make a case for further more robust studies to accurately characterize leptospirosis in South Sudan. COnsider my comments for the improvement of the manuscript.

Methods>Data analysis

Here there is reference to bivariable analysis being used to evaluate association between seroprevalence and various exposures. Was this meant to be univariable logistic regression where one variable is modeled against a single outcome? If not, kindly clarify the components of the bivariable logistic regression.

It is important to include the well-known limitation for cross-sectional studies which is the inability to determine causality/causal associations between variables and occurrence of leptospirosis. Consider recommending a longitudinal study for more robust understanding of these relationships.

Reviewer #2: The study on 'Seroprevalence and associated risk factors of Leptospira spp. among slaughterhouse workers in Western Bahr El Ghazal State, South Sudan' by Onafruo et al. provides information on the seroprevalence among slaughterhouse workers in South Sudan and associated risk factors. Medical survey data and samples from 250 workers across three slaughterhouses were collected. Blood samples were screened by MAT using a panel of 12 leptospiral serovars. The results are well presented and the conclusion drawn are justified. Here are a few minor suggestions/comments:

1. In the manuscript, wherever percentages are presented, please consider providing raw numbers as well.

2. Information in Table 1 and 3 can be combined.

3. Table 3 and 4 can be made 'supplementary figures'.

4. Screening of a subset of animals from the slaughterhouses for renal carriage and/or seroprevalence would make a good future study.

PLOS authors have the option to publish the peer review history of their article (what does this mean?). If published, this will include your full peer review and any attached files.

Reviewer #1: Yes: Dennis N. Makau

Reviewer #2: No
---

## [Decision Letter · Decision Letter 1]

16 Oct 2024

Dear Dr. Onafruo,

Thank you very much for submitting your manuscript "Seroprevalence and associated risk factors of Leptospira spp. among slaughterhouse workers in Western Bahr El Ghazal State, South Sudan" for consideration at PLOS Neglected Tropical Diseases. As with all papers reviewed by the journal, your manuscript was reviewed by members of the editorial board and by several independent reviewers. The reviewers appreciated the attention to an important topic. Based on the reviews, we are likely to accept this manuscript for publication, providing that you modify the manuscript according to the review recommendations. 

Sincerely,

Sreekumari Rajeev, BVSc, PhD

Guest Editor

Ana LTO Nascimento

Section Editor

Reviewer's Responses to Questions

**Key Review Criteria Required for Acceptance?**

**Methods**

-Are the objectives of the study clearly articulated with a clear testable hypothesis stated?

-Is the study design appropriate to address the stated objectives?

-Is the population clearly described and appropriate for the hypothesis being tested?

-Is the sample size sufficient to ensure adequate power to address the hypothesis being tested?

-Were correct statistical analysis used to support conclusions?

-Are there concerns about ethical or regulatory requirements being met?

Reviewer #1: No major comments in this section.

**Results**

-Does the analysis presented match the analysis plan?

-Are the results clearly and completely presented?

-Are the figures (Tables, Images) of sufficient quality for clarity?

Reviewer #1: No major comments in this section.

**Conclusions**

-Are the conclusions supported by the data presented?

-Are the limitations of analysis clearly described?

-Do the authors discuss how these data can be helpful to advance our understanding of the topic under study?

-Is public health relevance addressed?

Reviewer #1: Consider moving the limitations section of the conclusion to the last part of the discussion. The conclusion needs to be as brief as possible and only containing key take away and recommendations.

**Editorial and Data Presentation Modifications?**

Reviewer #1: The are several typographical and grammatical issues that need addressing, I have highlighted those in the general comments section below.

**Summary and General Comments**

Reviewer #1: L41: Check punctuation in this sentence after 'month'

L48: I think the more appropriate way of phrasing this sentence is to say that ‘people who wore aprons were 10.6 time more likely…..’

L50: Consider replacing 'from' with ‘by’ for grammatical correctness 

L94: Replace ‘are’ with ‘is’ for grammatical correctness

L112: It would be good to clarify why Wau Municipal was selected for this study. Was it for convenience?

L139: Replace ‘sample’ with ‘samples’ for grammatical correctness

L227: Did you mean ‘flaying’ instead of 'flying'?

L227: Was this variable considered as a proxy of wearing PPE or the act of wearing PPE itself? If it was the act itself then the extrapolation that it was a proxy should be brought up in the discussion and reported here as it were.

L235: Consider replacing 'from' with ‘by’ for correctness. Alternatively, this sentence can be phrased as ‘unit change in the number of carcasses handled'.

L270-273: Would the inability of including these workers in the study due to their busy schedule have introduced some bias in the study? Kindly consider commenting about this as a limitation of the study.

L285: Consider excluding 'were' for clarity in this sentence

PLOS authors have the option to publish the peer review history of their article (what does this mean?). If published, this will include your full peer review and any attached files.

Reviewer #1: Yes: Dennis Makau

Figure Files:

Data Requirements:

Reproducibility:

References

---

## [Decision Letter · Decision Letter 2]

14 Nov 2024

Dear Dr. Onafruo,

We are pleased to inform you that your manuscript 'Leptospira seroprevalence and associated risk factors among slaughterhouse workers in Western Bahr El Ghazal State, South Sudan' has been provisionally accepted for publication in PLOS Neglected Tropical Diseases.

Best regards,

Sreekumari Rajeev, BVSc, PhD

Guest Editor

Ana LTO Nascimento

Section Editor

Shaden Kamhawi

co-Editor-in-Chief

Paul Brindley

co-Editor-in-Chief

Reviewer's Responses to Questions

**Key Review Criteria Required for Acceptance?**

**Methods**

-Are the objectives of the study clearly articulated with a clear testable hypothesis stated?

-Is the study design appropriate to address the stated objectives?

-Is the population clearly described and appropriate for the hypothesis being tested?

-Is the sample size sufficient to ensure adequate power to address the hypothesis being tested?

-Were correct statistical analysis used to support conclusions?

-Are there concerns about ethical or regulatory requirements being met?

Reviewer #1: (No Response)

**Results**

-Does the analysis presented match the analysis plan?

-Are the results clearly and completely presented?

-Are the figures (Tables, Images) of sufficient quality for clarity?

Reviewer #1: (No Response)

**Conclusions**

-Are the conclusions supported by the data presented?

-Are the limitations of analysis clearly described?

-Do the authors discuss how these data can be helpful to advance our understanding of the topic under study?

-Is public health relevance addressed?

Reviewer #1: (No Response)

**Editorial and Data Presentation Modifications?**

Reviewer #1: (No Response)

**Summary and General Comments**

Reviewer #1: (No Response)

PLOS authors have the option to publish the peer review history of their article (what does this mean?). If published, this will include your full peer review and any attached files.

Reviewer #1: **Yes: **Dennis N. Makau

---

## [Editor Report · Acceptance letter]

24 Nov 2024

Dear Corresponding Author Onafruo,

We are delighted to inform you that your manuscript, "Leptospira seroprevalence and associated risk factors among slaughterhouse workers in Western Bahr El Ghazal State, South Sudan," has been formally accepted for publication in PLOS Neglected Tropical Diseases.

Best regards,

Shaden Kamhawi

co-Editor-in-Chief

Paul Brindley

co-Editor-in-Chief
